# Demonstration of SWIR Silicon-Based Photodetection by Using Thin ITO/Au/Au Nanoparticles/n-Si Structure

**DOI:** 10.3390/s22124536

**Published:** 2022-06-16

**Authors:** Xinxin Li, Zhen Deng, Ziguang Ma, Yang Jiang, Chunhua Du, Haiqiang Jia, Wenxin Wang, Hong Chen

**Affiliations:** 1Key Laboratory for Renewable Energy, Beijing Key Laboratory for New Energy Materials and Devices, Beijing National Laboratory for Condensed Matter Physics, Institute of Physics, Chinese Academy of Sciences, Beijing 100190, China; xinxin.li@iphy.ac.cn (X.L.); zgma@iphy.ac.cn (Z.M.); jiangyang@iphy.ac.cn (Y.J.); duchunhua@iphy.ac.cn (C.D.); mbe2@iphy.ac.cn (H.J.); wxwang@iphy.ac.cn (W.W.); hchen@iphy.ac.cn (H.C.); 2Center of Materials and Optoelectronics Engineering, University of Chinese Academy of Sciences, Beijing 100049, China; 3The Yangtze River Delta Physics Research Center, Liyang 213000, China; 4Songshan Lake Materials Laboratory, Dongguan 523808, China

**Keywords:** Au nanoparticle, dark current suppression, enhanced absorption, SWIR, photodetectors

## Abstract

Plasmonic photodetection based on the hot-electron generation in nanostructures is a promising strategy for sub-band detection due to the high conversion efficiencies; however, it is plagued with the high dark current. In this paper, we have demonstrated the plasmonic photodetection with dark current suppression to create a Si-based broadband photodetector with enhanced performance in the short-wavelength infrared (SWIR) region. By hybridizing a 3 nm Au layer with the spherical Au nanoparticles (NPs) formed by rapid thermal annealing (RTA) on Si substrate, a well-behaved ITO/Au/Au NPs/n-Si Schottky photodetector with suppressed dark current and enhanced absorption in the SWIR region is obtained. This optimized detector shows a broad detection beyond 1200 nm and a high responsivity of 22.82 mA/W at 1310 nm at −1 V, as well as a low dark current density on the order of 10^−5^ A/cm^2^. Such a Si-based plasmon-enhanced detector with desirable performance in dark current will be a promising strategy for realization of the high SNR detector while keeping fabrication costs low.

## 1. Introduction

Short-wavelength infrared (SWIR) photodetection has attracted tremendous attention for wide applications in spectroscopy [1], night surveillance [2,3], and telecommunication applications [4,5]. Generally, the Si-based photodetector (PD) is unsuitable for SWIR detection since there is little light absorption for wavelengths beyond 1100 nm, corresponding to its 1.12 eV bandgap. Consequently, the PDs with excellent performance for SWIR detection are mostly based on Ge-Si or III-V materials, which are suffering from expensive material costs, large noise characteristics, or poor integration with the complementary metal-oxide-semiconductor (CMOS) technologies [6,7,8]. Therefore, the development of low-cost Si-based PDs working in the SWIR band is of great importance, and many approaches have been proposed, including quantum dots (QDs) [9], black silicon technology [10,11,12], two-photon absorption (TPA) [13], internal photoemission (IPE) in Schottky junction [14,15,16], etc. However, the responsivities in SWIR of these devices are still significantly low and not sufficient to boost performance to be comparable with the commercial PDs in use. Recently, increasing interest has been paid to the plasmon-enhanced PD based on the surface plasmons (SPs)-assisted hot-electron generation mechanism, and it has made a great breakthrough in increasing responsivity [17,18,19,20,21]. In this configuration, the incident electromagnetic radiation from free space will be effectively harvested and confined to the nanoscale volume via the plasmons’ coherent oscillations of free hot carriers in metals, and then generate hot carriers via plasmon decay at high efficiencies [22]. Once these plasmon-induced hot carriers are transferred across the Schottky barrier (SB) at the metal–semiconductor interface, they will be collected as photocurrents to realize optical detection [23]. The outstanding light-trapping and electromagnetic-field-concentrating properties enable significant absorption enhancement, which is the basis for the effective improvement of PD responsivity [18,24,25,26]. For instance, the photocurrent detected at the Au/Si interface of the Si SP waveguide detector is much higher compared with that detected from the device without the diffraction structure (26 times for the multi-slit structure and 10 times for the disk array) [27]. The SPs excited from the random-sized Au nanoparticles (NPs) on Si surface result in effective absorption of sub-bandgap photons in silicon, and thus the responsivity of 2 mA/W and 600 µA/W are obtained at 1310 and 1550 nm, respectively [28]. However, few researchers pay much attention to the performance of dark current density. The signal-to-noise ratio (SNR) of a PD is determined by both the responsivity and dark current, and it has been demonstrated that the introduction of extra metal nanostructures will severely drag down the dark current performance of the device [28,29].

In the previous work, it was demonstrated that a metal layer with a high work function on the semiconductor surface can effectively increase the SB height, leading to a reduced dark current of the detector [30,31,32]. Here, a 3 nm Au layer is covered on the Au NPs formed by rapid thermal annealing (RTA) on the Si surface to utilize the advantages of both the high SB from the Au layer and the SPs’ excitation in NPs. The experimental results show that the ITO/Au/Au NPs/n-Si PD has a suppressed dark current density on an order of 10^−5^ A/cm^2^, with a high responsivity beyond 1200 nm through the plasmons-assisted hot-electron generation mechanism. This trade-off result of a suppressed dark current and an enhanced absorption meets the development tendency of PDs towards low-cost and high SNR, which will be a high-profile strategy to realize high-performance Si-based SWIR PDs.

## 2. Experiment

### 2.1. Device Fabrication

Figure 1 shows the schematic view of the structure of ITO/Au/Au NPs/n-Si PDs. The area (0.02 cm^2^) of the top electrode, i.e., the light-sensitive region, was defined by the ultraviolet lithography on a commercial epi-ready n-type Si wafer (0.1−1 Ω·cm, <100>). To form the Au NPs on the Si surface for SPs’ excitation, a nano-thickness Au layer was firstly deposited in the defined area. Since the lower deposition energy from the thermal evaporation (RE) [33] facilitated the formation of uniform Au NPs after subsequent RTA, this layer would be prepared by the RE equipment (ULVAC ei-501z, ULVAC, Methuen, MA, USA) with a deposition rate of 0.1 Å/s. During the RTA process, the temperature was raised from room temperature to the target temperature with a heating rate of 15 °C/s and held for the required time, then naturally cooled down to room temperature. Besides, the whole annealing process was carried out in N_2_ gas protection. The morphology of Au NPs was tailored by controlling the thickness of the annealed Au layer and the condition of the followed RTA, including the temperature and time [34,35]. The other thin Au layer covered on the Au NPs in Figure 1 was prepared by electron beam evaporation (EBE, Ohmiker-50B, M & R Technology, Noida, India) with a deposition rate of 0.17 Å/s to form an intimate Au–Si contact for a high SB to suppress the dark current. After that, a 100 nm capping layer of the transparent conductive glass (ITO) was deposited as the conductive electrode by a double-chamber magnetron sputtering system (Shenyang Defeng Technology) under Ar/O_2_ atmosphere. On the other hand, the electrode for ohmic contact on the back of Si was realized by 300 nm Al using Ohmiker-50B. During the preparation of PDs, the oxide layer on the Si surface would be removed with dilute HF solution (H_2_O:HF = 10:1 for 1 min) before all metal deposition to improve the contacts. Figure 1 also shows the mechanism of performance improvement of such a structure. As we can see, the incident radiation with energy larger than the barrier height excites the SPs with the help of Au NPs, followed by the hot electrons’ generation through the non-radiative decay. These hot electrons will be transported to the M/S interface before thermalization and then injected into the conduction band to form the photocurrent. On one hand, both the outstanding light-trapping and electromagnetic-field-concentrating properties of SPs enable the significant absorption enhancement in the SWIR band. On the other hand, a thin Au layer has been introduced to effectively increase the SB height, leading to a reduced dark current of the detector.

### 2.2. Characterization and Analysis

The morphology of Au NPs on the Si substrate was directly observed on a scanning electron microscope (SEM, SUS5500) with an operating voltage of 15 kV. To study the size information of Au NPs prepared from different conditions more scientifically, the numbers and their area duty cycles (ADCs) of NPs with different sizes were statistically counted with the help of Image-Pro Plus image analysis software. As for the photoelectric performance of the PD, the current–voltage characteristics were recorded using a Keithley 2400 semiconductor parameter analyzer under dark and laser radiation (1310 nm, the effective input power was about 1 mW). Furthermore, the spectral characterization was measured by Fourier transform infrared spectroscopy (FTIR, Bruker Vertex70). During the measurement of the optical signal, the PDs were investigated at normal incidence with the same distance from the light source to ensure the consistency of test conditions

## 3. Results

To obtain Au NPs that can enhance the absorption in the SWIR band with SPs’ excitation, the sizes of the required Au NPs must be optimized first. Since the nanostructures formed by RTA have a size dispersion [35,36], resulting in a good tolerance to the wavelength range of enhanced absorption, we preliminarily speculated that the required size of Au NPs (for 1200−1700 nm) is around 110 nm by reference to the reported results [20,28]. Figure 2a−f are the SEM images of samples with the as-deposited Au thickness of 5 and 10 nm after being annealed at different temperatures for 10 min. They are the: (a) 5 nm Au layer annealed at 200 °C, (b) 5 nm Au layer annealed at 450 °C, (c) 5 nm Au layer annealed at 600 °C, (d) 10 nm Au layer annealed at 200 °C, (e) 10 nm Au layer annealed at 450 °C, and (f) 10 nm Au layer annealed at 600 °C, referred to as S1−S6, respectively. It is clearly seen that, by controlling the annealing temperature, the typical evolution of the Au nanostructures on Si can be observed [37]. The samples annealed at 200 °C had fragmented and sharp structures with uneven distribution (Figure 2a,d). With the increase of the annealing temperature to 450 °C, the fragmented structures gradually merged into the isolated sphere-like islands with a narrow size distribution (Figure 2b,e). When the temperature further increased to 600 °C, some of the small-sized structures continued to merge into larger structures, resulting in a broad size distribution (Figure 2c,f). In addition, the thicker annealed Au layer of 10 nm formed larger-sized structures at the same RTA condition compared to the thinner Au layer of 5 nm, which is consistent with the previous studies [34,38]. According to the morphology of Au NPs shown in the above SEM images, the sizes of Au NPs in S2 were distributed around 100 nm, which might meet the requirement for efficient absorption in 1200−1700 nm considering the sizes and uniformity. 

For the sake of exploring the actual absorption of Au NPs of S2 in the range of 1200–1700 nm, Si-based PDs with two structures were prepared: (1) ITO (100 nm)/Au NPs (S2)/n-Si PD (target PD) and (2) the reference ITO (100 nm)/n-Si PD, referred to as PD-S2 and PD-ITO, respectively. The measured photoresponsivity spectra are plotted in Figure 2g. Compared with PD-ITO, PD-S2 showed a significantly enhanced response with a responsivity of 16.44 mA/W at 1300 nm due to the Au NPs’ insertion. The value is much higher than the responsivity of 2 mA/W from the same structure reported in the literature [28], which is mainly because of the stronger enhancement from the well-distributed and irregularly shaped NPs in this work. These results confirm that Au NPs prepared by annealing a 5 nm Au layer on Si at 450 °C for 10 min can be used for Si-based Schottky detectors to improve the response in the SWIR band. In the J−V semi-log plot from –1 to 0 V (Figure 2h), PD-S2 showed a smaller dark current density than PD-ITO without Au NPs, which is mainly attributed to the more ideal Au–Si interface in this study. The dark current suppression from the SB of Au-Si was greater than the dark current increment from the generation–recombination centers near the interface caused by the Au NPs. However, it should be noted that both PD-S2 and PD-ITO have a dark current density of over 1 × 10^−3^ A/cm^2^. Such a high-level dark current is detrimental to the sensitivity, making it difficult for the PD to be utilized in practical applications.

Based on the above structure, a 2 nm Au layer was introduced to cover the exposed Si surface, except for Au NP areas, so as to improve the effective SB height and suppress the dark current of the Au NPs-decorated Si-based SWIR PD. Meanwhile, the thickness of the annealed Au layer will be fine-tuned around 5 nm to further enhance the absorption in the range of 1200−1700 nm. Therefore, the Si-based PDs with different thicknesses of the annealed Au layer were prepared: (1) ITO (100 nm)/Au (2 nm)/Au NPs (3.5 nm-450-10)/n-Si, (2) ITO (100 nm)/Au (2 nm)/Au NPs (5 nm-450-10)/n-Si, and (3) ITO (100 nm)/Au (2 nm)/Au NPs (6.5 nm-450-10)/n-Si, referred to as PD-S7-2, PD-S8-2, and PD-S9-2, respectively. The morphology of Au NPs formed by annealing the Au layers of 3.5, 5, and 6.5 nm at 450 °C for 10 min and coating a 2 nm Au layer is shown in Figure 3a−c, referred to as S7-2, S8-2, and S9-2. The upper part of the figure shows the morphology under the magnification of ×10 K, while the lower part shows the more detailed morphology under the magnification of ×35 K. As shown in the SEM images, the Au layers of 5 ± 1.5 nm formed the discrete spherical structures with uniform size distribution and regular shape after annealing at 450 °C for 10 min. With the increase of the Au layer thickness, the sizes of the NPs tended to be larger, which is consistent with that shown in Figure 2a−f. Moreover, it can be seen that coating the 2 nm Au layer showed little effect on the morphology of the NPs. On the contrary, the Au layer increased the conductivity of the sample and weakened the influence of charge accumulation on the sample surface, resulting in a higher image resolution. In addition, the morphological characteristics of Au NPs prepared under the same conditions shown in Figure 2b and Figure 3b are almost the same, which suggests the good repeatability of our preparation. To study the distribution information of Au NPs more scientifically, the numbers and the area duty cycles (ADCs) of NPs with different sizes were statistically counted in Figure 3d,e. The size distribution of the NPs formed with the 3.5 nm Au layer in Figure 3d was narrow with a near-Gaussian distribution, and the main sizes were around 70−90 nm. Those formed with a 5 nm Au layer were wider with a more even distribution, concentrated in 70−120 nm. When the annealed Au layer increased to 6.5 nm, the NPs with larger sizes (>160 nm) began to appear, resulting from more Au matter used for reshaping, but not constituting the majority. Since the islands with different sizes occupied different areas, resulting in the different Si–Au contact, the ADCs of sizes are presented in Figure 3e. It can be seen that their ADCs were more consistent with the Gaussian distribution, and the main ADCs of the islands were concentrated in 70−90, 90−140, and 140−260 nm for S7-2, S8-2, and S9-2, respectively. This distribution information will affect the overall device performance, including dark current and responsivity.

Figure 3f,g exhibit the J−V curves of the three devices under the dark condition from −1 to 1 V, as well as that of PD-S8 (the same structure with PD-S2) for reference. Under the linear scale in Figure 3f, the devices showed different turn-on voltages and could be extracted as 0.15, 0.20, 0.22, and 0.24 V for PD-S8, PDS7-2, PD-S8-2, and PD-S9-2, respectively. The comparison of dark current densities is exhibited more directly in the J−V semi-log plot under the reverse bias in Figure 3g. It should be noted that the dark current of PD-S8 was significantly suppressed from 2.74 × 10^−5^ to 2.24 × 10^−6^ A at −1 V after being covered with a layer of 2 nm Au (PD-S8-2). Although there was a small difference in the dark current density between PDs with the 2 nm coating Au layer, which was partly caused by the different sized Au NPs, they all showed much lower dark currents compared to PD-S8. Considering the similar structures of these PDs, this general reduction of dark current can be attributed to the 2 nm coating Au layer. Moreover, the turn-on voltages (V_on_), the Φ_B_s extracted with the method reported in [39], and the dark currents (I_d_) at −1 V of the above devices are summarized in Table 1. The dark current density values under the reverse bias correspond well to the turn-on voltages extracted from Figure 3f; that is, a higher turn-on voltage indicates a higher barrier height, resulting in a lower dark current.

To analyze the mechanism of the dark current suppression from coating a thin Au layer on the Au NPs in PD (PD-S8-2), the cross-sectional SEM image of the device was measured and is displayed in Figure 4a. Each layer of the structure can be seen except for the 2 nm Au coating layer, limited by the resolution of SEM equipment. The thickness of ITO deposited on Si substrate was about 100 nm, with recognizable spherical shapes embedded therein. The regular spherical shape surrounded by the red-dotted line can be easily distinguished, and it is the holes after the Au NPs peeled off due to sample preparation, which suggests the Au nanoparticle shape. In addition, a schematic diagram of Au atoms’ deposition path to the NPs is shown. The spherical structure of the NPs forms a spatial space in contact with Si. When Au atoms come from the source by evaporation, the region below NPs cannot be covered by Au atoms because of the blocking from the NPs, resulting in the Au-free blind region. Figure 4b shows the energy band diagram of the ITO/Au/Au NPs/Si structure. Since the work function of ITO (≈4.5 eV) is smaller than that of Au [22], an energy well in Au will be formed after inserting Au material between ITO and n-Si, resulting in a higher SB of Au/n-Si contact, as shown in the energy band diagram. The actual contact of the Au-free blind region is a contact of ITO/n-Si, the barrier of which is much lower than that of Au/n-Si contact, and this may lead to a decrease of the effective SB height of the detector and weaken the suppression of the dark current. The analysis is in good agreement with the dark current performance of different structures presented in Table 1. The NP with a larger size in PD-S9-2 deviated more from the ideal spherical structure (Figure 3c), creating a larger contact of Au NP/n-Si. Consequently, the proportion of the total area blocking Au atoms was smaller, bringing about the higher SB height and lower dark current after covering the same 2 nm Au layer.

Another PD with similar structural parameters to PD-S8-2 but increasing the coating Au thickness to 3 nm was prepared to further confirm our speculation, which is referred to as PD-S8-3. The dark currents and the photocurrents measured at 1310 nm of PD-S8-2 and PD-S8-3 from −1 to 0.3 V are shown in Figure 5a. It resulted that the dark current was further suppressed from 2.24 × 10^−6^ to 8.8 × 10^−7^ A at −1 V as the Au coating layer increased from 2 to 3 nm. This is mainly attributed to the enhanced side-diffusion caused by the thicker Au layer deposition [40], which effectively reduced the Au-free blind area, thus enlarging the Au–Si contact and decreasing the dark current. In addition, the rectification ratios (the ratio of currents at ±1 V) for PD-ITO, PD-S8, PD-S8-2, and PD-S8-3 were 1.1 × 10^3^, 2.0 × 10^3^, 2.0 × 10^4^, and 2.0 × 10^5^, respectively, which exhibited a similar trend as the dark currents. The main reason is that the currents under forward bias showed little difference for detectors with similar structures due to the reliable series resistances. Moreover, the results of the photocurrent test indicated that the response of PD-S8-3 was also improved at 1310 nm. PD-S8-3 exhibited a higher responsivity of 22.82 mA/W compared to the 16.92 mA/W of PD-S8-2, which was mainly due to the enhanced absorption from the adjustment of Au NPs by a thicker Au layer. The performance improvement of PD-S8-3 compared to PD-ITO, PD-S8, and PD-S8-2 is shown in Figure 5b, including the dark current and the responsivity at 1310 nm at −1 V. As shown, the introduction of the Au coating layer is essential for dark current suppression. The PD-S8 without an Au coating layer exhibited a high dark current of 2.74 × 10^−5^ A for the reverse bias of 1 V. On the contrary, with the improvement of Au–Si contact from the Au coating layer, the effective SB of the device increased, leading to a decrease of the dark current to 2.24 × 10^−6^ A for PD-S8-2 and 8.83 × 10^−7^ A for PD-S8-3. In terms of the responsivity, the absorption was improved with the SPs’ excitation when Au NPs were involved, resulting in an increased responsivity from 0.50 to 16.44 mA/W. Besides, the amplitude of enhancement can be adjusted to a high level (22.82 mA/W) by further optimizing the morphology of Au NPs. As a result, a Si-based SWIR PD with improved responsivity and a suppressed dark current was obtained with the optimized ITO/Au/Au NPs/Si structure.

The photocurrent-to-dark current ratios (PDRs) of the PDs, defined as the ratio of the photocurrent under 1310 nm to the dark current, are presented in Figure 5c as a function of bias in a semi-log scale. Overall, all the PDs exhibited a similar trend of PDR when the applied voltage changed. The PDR was relatively stable with the bias, but it fluctuated greatly when it was near 0 V and the turn-on voltage, which is attributed to the sharp changes of the dark current and the photocurrent near both voltages, respectively. The PDR values at 0 V were 1, 32, 430, and 1009 for PD-ITO, PD-S8, PD-S8-2, and PD-S8-3, respectively. In addition, the PDRs of the PDs at 1310 nm for the reverse bias of 1 V are provided in Figure 5d. As it is shown, the PDR of the PD-S8-3 was greatly improved due to both the increase of absorption and the suppression of the dark current, representing a higher SNR of the device. Comparing Figure 5d with Figure 5b, it is important to note that a large dark current will severely drag down the PDR of the device, and thus the suppression of the dark current is particularly important to have a high SNR. Besides, it is more practical to improve the SNR through the dark current suppression, since the dark current presents the exponential relationship with the SB in a Schottky detector [41], which is one of the bases of this work. Figure 5e shows the responsivity spectrum of PD-S8-3 with the highest response and the best PDR obtained in this study, as well as that of PD-ITO without Au material and PD-5Au (ITO (100 nm)/Au (5 nm)/n-Si) without NPs for comparison. It can be seen that the response of the PD-S8-3 showed a significant improvement in the broadband with the help of the appropriate Au NPs instead of the Au material, especially in the wavelengths from 1200 to 1500 nm. This result confirms that the enhanced absorption in the ITO/Au/Au NPs/Si structure comes from the SP-assisted hot-electron generation mechanism in Au NPs. The response of PD-S8-3 at 1200 nm was about 45 mA/W, which is nearly 10 times higher than the 3.5 mA/W of Si PD with random-sized plasma nano-antennas reported in [28]. In addition, the internal quantum efficiency (IQE) spectra as a function of wavelength were also observed, as shown in Figure 5f, which showed the same trend as the responsivity spectra in Figure 5e. As a result, a Si-based SWIR PD with improved responsivity beyond 1200 nm and a suppressed dark current density of 4.4 × 10^−5^ A/cm^2^ at −1 V were obtained with the optimized ITO/Au/Au NPs/Si structure.

The properties of PD-S8-3 are summarized in Table 2, together with other Si-based plasmon-enhanced (Au) PDs used for the telecommunication wavelength for comparison. In comparison with the Au/Si plasmon-enhanced PDs with complicated designs, such as the addition of graphene or using precise structures formed with electron beam lithography in Table 2, this ITO/Au/Au NPs/n-Si PD with simple processing exhibited a higher SB height, resulting in a lower dark current density [20,26,42,43]. We have also investigated the detectors with random-sized Au NPs formed by RTA to support the SPs’ excitation, such as the structure of ITO/Au NPs/n-Si and ITO/Au/n-pyramid Si [28,38]. Apart from the relatively low dark current density resulting from the thin Au insertion layer, the PD-S8-3 also showed a higher responsivity in the SWIR band due to the optimized Au NPs. 

## 4. Conclusions

In conclusion, we have demonstrated a Si-based PD with a suppressed dark current density as well as enhanced absorption operating in a broad sub-bandgap region. Regular spherical structured Au NPs with controllable distribution have been fabricated by the low-cost RTA method to realize the responsivity increment with the SP-assisted hot-electron generation mechanism, while a 3 nm-thin Au layer was used to improve the effective SB and reduce the dark current. This hybrid structure of ITO/Au/Au NPs/n-Si PDs with a low dark current density of 4.4 × 10^−5^ A/cm^2^ showed detection beyond the 1200 nm wavelength and a responsivity of 22.82 mA/W at 1310 nm at −1 V. We believe that this demonstration of dark current suppression with plasmonic hot carrier generation on a Si-based detector is a step forward in meeting the requirements for a high SNR with extreme simplicity of fabricating nanoscale structures.

## Figures and Tables

**Figure 1 sensors-22-04536-f001:**
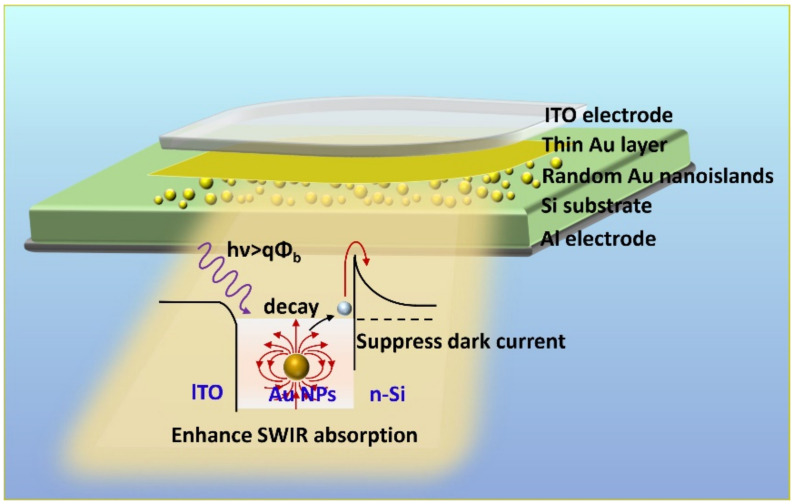
The schematic view of the structure of the ITO/Au/Au NPs/n-Si PD and the mechanism of performance improvement.

**Figure 2 sensors-22-04536-f002:**
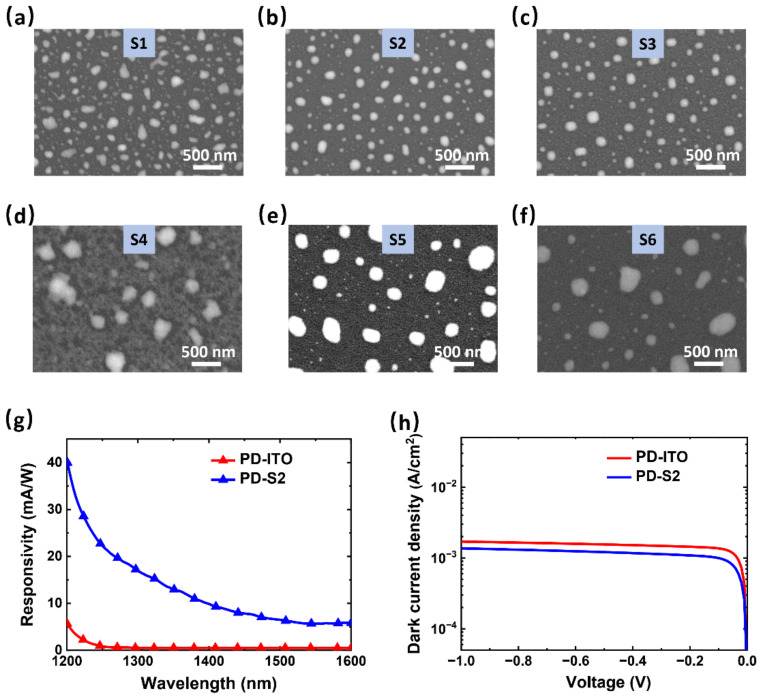
(**a**–**f**) SEM images of Au NPs’ distribution annealed from Au layers on Si substrate with thicknesses of (**a**) 5 nm (at 200 °C), (**b**) 5 nm (at 450 °C), (**c**) 5 nm (at 600 °C), (**d**) 10 nm (at 200 °C), (**e**) 10 nm (at 450 °C), and (**f**) 10 nm (at 600 °C) for 10 min, respectively. (**g**) The responsivity spectra of PD-S2 and PD-ITO. (**h**) The dark current densities in semi-log scale of PD-S2 and PD-ITO, from −1 to 0 V.

**Figure 3 sensors-22-04536-f003:**
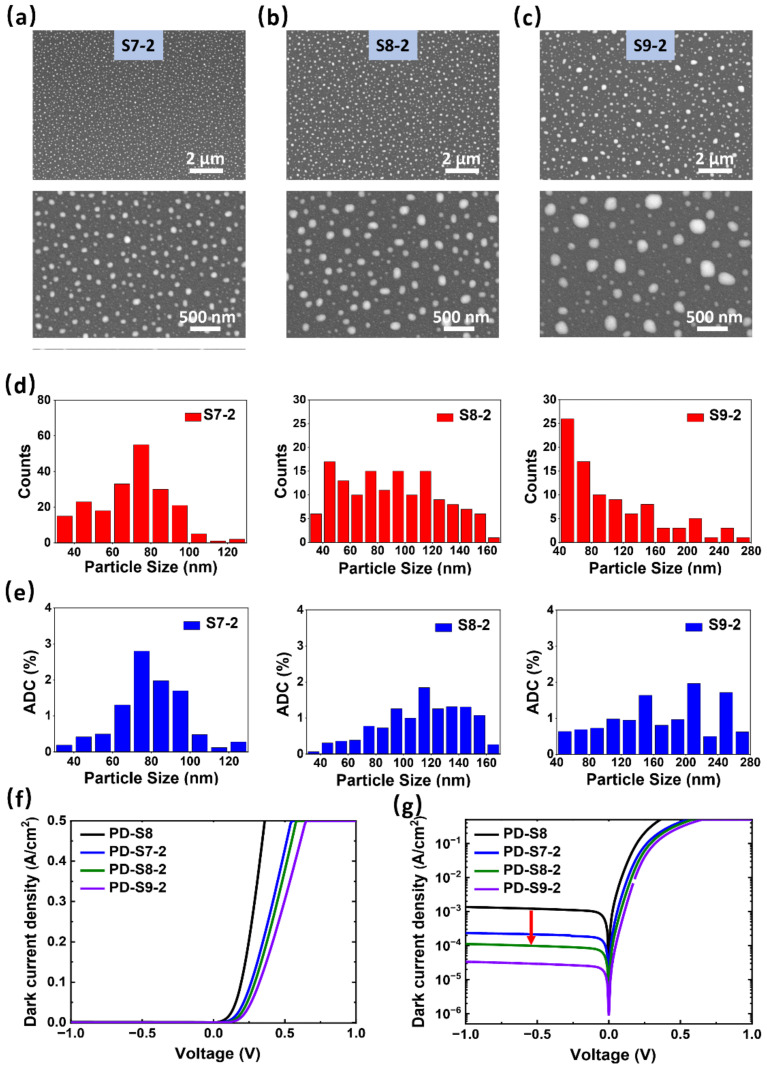
SEM images of Au NPs’ distribution with 2 nm Au layers capped on. The thicknesses of the Au layer to be annealed at 450 °C for 10 min were (**a**) 3.5, (**b**) 5, and (**c**) 6.5 nm, respectively. Histograms of NPs’ characteristics exported from the figure (**a**): (**d**) size and (**e**) area duty cycle (ADC). The dark current densities of PDs with different Au NPs at −1 to 1 V in (**f**) linear scale and (**g**) semi-log scale.

**Figure 4 sensors-22-04536-f004:**
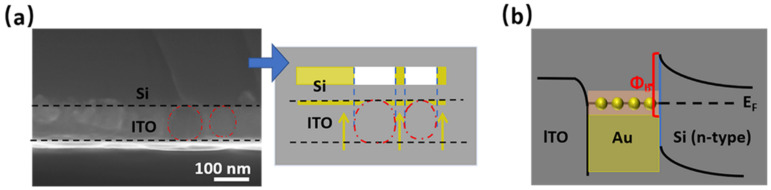
(**a**) Cross-sectional SEM image (left) of the ITO/Au/Au NPs/Si structure (PD-S8-2) with a schematic diagram of the Au atoms’ deposition path to the NPs (right). The area surrounded by the red-dotted line embedded in ITO is the hole after the Au NP fell off due to the sample preparation. These nearly spherical NPs block the subsequent deposited Au atoms, resulting in the blind regions covered with ITO instead of Au at the bottom of the NPs. (**b**) Energy band diagram of the ITO/Au/Au NPs/Si structure.

**Figure 5 sensors-22-04536-f005:**
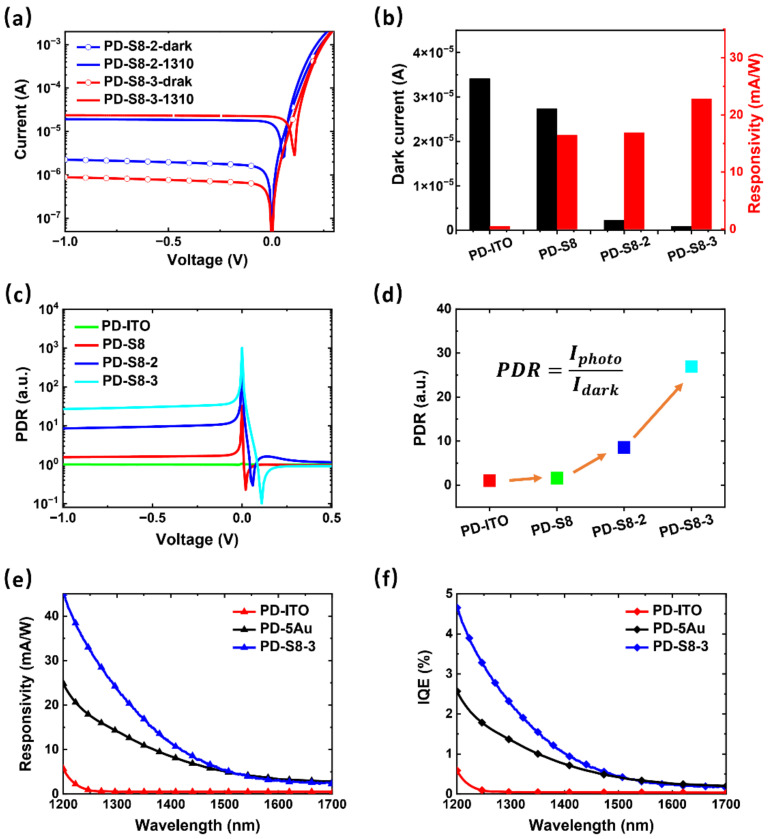
(**a**) The dark currents and photocurrents at 1310 nm of PD-S8-2 and PD-S8-3 from −1 to 0.3 V. (**b**) The comparison of PDs with different structures in dark current and the responsivity at 1310 nm at −1 V. (**c**) The PDRs of the PDs at 1310 nm as a function of bias. (**d**) The PDRs of the PDs at 1310 nm for the reverse bias of 1 V. (**e**) The responsivity spectra and (**f**) the internal quantum efficiency (IQE) spectra of PD-ITO, PD-5Au, and PD-S8-3.

**Table 1 sensors-22-04536-t001:** V_on_, Φ_B_, and I_d_ at −1 V of PD-S8, PD-S7-2, PD-S8-2, and PD-S9-2.

PDs	PD-S8	PD-S7-2	PD-S8-2	PD-S9-2
V_on_ (V)	0.15	0.20	0.22	0.24
Φ_b_ (eV)	0.57	0.61	0.64	0.67
I_d_ (A)	2.74 × 10^−5^	4.72 × 10^−6^	2.24 × 10^−6^	6.73 × 10^−7^

**Table 2 sensors-22-04536-t002:** Performance comparison for Si-based plasmon-enhanced (Au) PDs.

Configuration	Special Design	SBH (eV)	Dark Current Density (A/cm^2^) (at −1 V)	Responsivity (mA/W) (1310 nm)	Refs.
Au/p-Si	waveguide	0.31	6	13.3 at 0.1 V	[20]
Au/graphene/p-Si	graphene + waveguide	0.34	1.3	85 at –1 V (1550 nm)	[42]
Au/p-Si	nanograting	0.32	48.0	14.5 at 0 V (1550 nm)	[26]
Au/n-Si	Au antenna + Si nanowire array	0.46		∼36 at –1 V	[43]
ITO/Au/n-pyramid Si	Au NPs + pyramid-Si		2 × 10^–5^	∼5 at 0 V	[38]
ITO/Au NPs/n-Si	Au NPs		5 × 10^–3^	2 at 0 V	[28]
PD-S8-3	Au + Au NPs	0.66	4.4 × 10^–5^	21.7 at 0 V 22.82 at –1 V	

## Data Availability

Not applicable.

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
