# Peer review of "Demonstration of SWIR Silicon-Based Photodetection by Using Thin ITO/Au/Au Nanoparticles/n-Si Structure"

_sensors, 2022, doi:10.3390/s22124536_

Round 1

Reviewer 1 Report

The manuscript entitled “Demonstration of Dark Current Suppression with Plasmonic Hot Carrier Generation for High-Performance SWIR Silicon- Based Photodetection by using thin ITO/Au/Au nanoparticles/n-Si structure" had studied the characteristics of plasmonic IR photodetector using ITO/Au/AuNPs/n-Si structure. Minor revision requires before acceptance with suitable corrections

1. Compare the photocurrent obtained in the present study with the literature values and discuss to evaluate the novelty of the present work.

2. Instead of writing sample name Au NPs in 5 nm-450-10 like this, it is better to  mention sample name like S1, S2.

3. Absorption spectra of PD-5Au-450-10 and PD-ITO samples would have been measured and compared.

4. Author need to explain charge transfer mechanism using proper band diagram.

Reviewer 2 Report

  • I suggest shortening the title to reflect the main contribution of the authors in this research work. I suggest it be just two lines instead of four lines. For example: “Demonstration of SWIR Silicon-Based Photodetection by using thin ITO/Au/Au nanoparticles/n-Si structure”.
  • In Fig. 4, the structure details cannot be easily resolved or distinguished clearly. I was expecting a clear high resolution figure like those presented in the authors’ previous work [14] (e.g. Fig. 2 and Fig. 7). Also, I suggest adding dotted lines to outline different areas of the photodetector structure in the figure. In addition, the authors should add more explanations and comments within the text on this figure.
  • The authors’ claims of high performance should be compared with their previous work in Ref. [14], which I believe has better values of dark current and responsivity.
  • How much is the rectification ratio for different photodetectors designs presented here?
  • In Fig. 1, I believe the authors should replace NIR with SWIR as it is the claimed wavelength range of their photodetector.
  • The vertical axis in Fig. 3-e should be labeled ADC instead of ADR.

Reviewer 3 Report

In this paper, “Demonstration of dark current suppression with plasmonic hot carrier generation for high-performance SWIR silicon-based photodetection by using thin ITO/Au/Au nanoparticles/n-Si structure”, the authors propose plasmonic photodetection with dark current suppression to create a Si-based broadband photodetector with enhanced performance in SWIR. Based on the obtained results, the authors claimed that such a Si-based plasmon-enhanced detector with desirable performance in dark current will be a promising strategy for realization of high SNR detector while keeping fabrication costs low. Overall, this manuscript has a strong potential for another review round after applying the issues and addressing the shortcomings listed below:

1-The authors should polish/revise some grammatical mistakes and typos along the manuscript. I invite the authors to read their manuscript carefully and make the required changes where necessary.

2-In the Introduction section, while discussing recent developments in the plasmonic photodetectors, more work should also be considered and cited to give a more general view to the possible readers of the work.

3-What will be the possible effect of using p-Si on the overall photodetection performance? Please explain.

4-Please increase font size of the text in Figures 3d-3g. Do the same for Figure 5 (all panels).

5-Please plot IQE as a function of wavelength.

6-Please plot photodetectivity of the device as a function of bias.

Round 2

Reviewer 2 Report

- I recommend adding the rectification ratios mentioned by the authors in their reply to the manuscript text.

- I recommend putting the two sub-figures in figure 4-a to become side by side.
